# Fecal microbiome composition and diversity of cryopreserved canine stool at different duration and storage conditions

**Patrick Barko, Julie Nguyen-Edquilang[ORCID], David A. Williams, Arnon Gal[ORCID]***

Department of Veterinary Clinical Medicine, College of Veterinary Medicine, University of Illinois at Urbana-Champaign, Urbana, IL, United States of America

* agal2@illinois.edu

**Data Availability Statement:** Data and R code necessary to replicate the analyses are in our GitHub repository (https://github.com/pcbarko/K9_FMT_Storage_Stability). The long-read 16S-rRNA

## Abstract

Fresh-frozen stool banks intended for humans with gastrointestinal and metabolic disorders have been recently established and there are ongoing efforts to establish the first veterinary fresh-frozen stool bank. Fresh frozen stored feces provide an advantage of increased availability and accessibility to high-quality optimal donor fecal material. The stability of frozen canine feces regarding fecal microbiome composition and diversity has not been reported in dogs, providing the basis for this study. We hypothesized that fecal microbial composition and diversity of healthy dogs would remain stable when stored at -20°C and -80°C for up to 12 months compared to baseline samples evaluated before freezing. Stool samples were collected from 20 apparently healthy dogs, manually homogenized, cryopreserved in 20% glycerol and aliquoted, frozen in liquid nitrogen and stored at -20°C or -80°C for 3, 6, 9, and 12 months. At baseline and after period of storage, aliquots were thawed and treated with propidium monoazide before fecal DNA extraction. Following long-read 16S-rRNA amplicon sequencing, bacterial community composition and diversity were compared among treatment groups. We demonstrated that fresh-frozen canine stools collected from 20 apparently healthy dogs could be stored for up to 12 months at -80°C with minimal change in microbial community composition and diversity and that storage at -80°C is superior to storage at -20°C. We also found that differences between dogs had the largest effect on community composition and diversity. Relative abundances of certain bacterial taxa, including those known to be short-chain fatty acid producers, varied significantly with specific storage temperatures and duration. Further work is required to ascertain whether fecal donor material that differs in bacterial community composition and diversity across storage conditions and duration could lead to differences in clinical efficacy for specific clinical indications of fecal microbiota transplantation.

## Introduction

The gut bacterial microbiome comprises many different genera, most of which are commensal or mutualistic, that participate in regulating host gut mucosal homeostasis, immunity, nutrient

amplicon sequences are deposited in the National Center for Biotechnology Information (NCBI) BioProject repository (https://www.ncbi.nlm.nih.gov/bioproject/PRJNA1001120) under the accession number PRJNA1001120. Other data presented in this study are available on request from the corresponding author.

**Funding:** Arnon Gal received funding from the American Kennel Club Canine Health Foundation Inc (https://www.akcchf.org; CHF Grant No. 02900). The funders had no role in study design, data collection and analysis, decision to publish, or preparation of the manuscript.

**Competing interests:** The authors have declared that no competing interests exist.

digestion and absorption, endocrine and neuroendocrine signaling, and many aspects of metabolism. Perturbations of the gut microbiome can affect the well-being of the host and enteric microbiota dysbiosis has been associated with many gastrointestinal, systemic, and metabolic diseases. The ability to understand the dynamics of host-microbiome interactions stem from technological advancements in the depth of DNA sequencing and related bioinformatic pipelines.

Fecal microbiota transplantation (FMT) is a medical procedure of emerging importance whereby feces from a healthy donor are administered to a diseased recipient to restore the composition and diversity of commensal microbiota. FMT is an FDA-approved therapy for recurrent *Clostridioides difficile*-associated colitis in humans (https://www.fda.gov/vaccines-blood-biologics/vaccines/rebyota; https://www.fda.gov/vaccines-blood-biologics/vowst) [1, 2]. Emerging evidence suggests FMT may also be effective for a variety of other gastrointestinal and systemic diseases in humans and other animals, including dogs [3, 4]. The hypothetical mechanism of action of FMT is dependent upon the transfer of viable bacteria from a healthy animal into a diseased recipient. There is also evidence indicating that cellular components from dead bacteria can also result in biological effects after application of FMT [5]. Traditional genomic DNA sequencing methods cannot distinguish DNA derived from bacteria with intact cell membranes from free DNA that leaked from bacteria with compromised cell membranes. Propidium monoazide (PMA) dye is a photoreactive chemical that preferentially binds double-stranded DNA, rendering it inert as a substrate for downstream PCR [6]. Propidium monoazide binds free DNA in solution and infiltrates bacteria with damaged cell membranes but cannot penetrate bacteria with intact cell membranes. Thus, DNA which remains intact for downstream sequencing in a PMA-treated sample is derived only from bacteria with intact cell membranes. This specificity allows us to assess changes exclusive to bacterial populations with uncompromised cell membranes, though it is important to note that bacteria with intact cell membranes are not necessarily viable or functional.

In recent years, fresh-frozen stool banks have been established in the USA, UK, Australia, and the Netherlands to facilitate large-scale implementation of FMT in human patients with recurrent *Clostridioides difficile*-associated colitis infections [7–11]. Compared to fresh stools, fresh frozen stored feces provide an advantage of increased availability and accessibility to high-quality optimal donor fecal material. The stability of frozen canine feces with regard to fecal microbiome composition and diversity has not been reported in dogs. As FMT has been a target of several recent investigations [12–20], there is an immediate need to assess microbiome composition and diversity across different storage conditions and durations, thus providing the basis for this study.

We hypothesized that fecal microbial composition and diversity of healthy dogs would remain stable when stored at -20˚C and -80˚C for up to 12 months compared to baseline samples evaluated before freezing. The primary objectives of this study were to investigate the effect of storage time and temperature on fecal microbiota composition and diversity. Specifically, we aimed to determine if storage at temperatures of -20˚C or -80˚C for up to 12 months will result in significant changes in microbial composition and diversity compared to baseline (fresh) samples. The secondary objectives of this study were to determine if microbial communities from different donor dogs are differentially stable, and which features are associated with fecal sample stability during storage.

## Methods

We prospectively collected stool samples from 20 apparently healthy staff- and student-owned dogs following receipt of each owner's written consent to participate in the study

(demographic information of the dogs is in Table 1). The dogs had no history of gastrointestinal dysfunction and administration of antibiotics, immunomodulatory, or gastroprotectant drugs 6 months before enrollment. We instructed owners to bring their dog's fresh morning stool within 2 hours of defecation.

## Sample preparation

Stool samples were manually homogenized in cold sterile 1×PBS, filtered through a metal sieve, and centrifuged at 600× *g* for 10 minutes at 4°C, discarding the supernatant. Each fecal pellet was resuspended in 20% glycerol in PBS for cryopreservation. Aliquots of each resuspended stool sample were transferred to sterile cryogenic vials, snap-frozen in liquid nitrogen, and stored at -20°C or -80°C for 3, 6, 9, and 12 months. Additional aliquots of resuspended stool in 20% glycerol in PBS from each dog were used as a baseline samples (i.e., not frozen), and their DNA was extracted immediately following treatment with PMA. At 3-month, 6-month, 9-month, and 12-month, the frozen aliquots were thawed on a dry heating block at 22°C for approximately 10 minutes and immediately treated with PMA before DNA extraction. To understand the effect of PMA treatment on fecal microbiome composition, feces samples from 10 of the dogs were pooled together in equal quantities and were aliquoted to 6 samples to serve as lysis/PMA controls. Fresh samples, and those where lysis was induced by exposure to heat (100°C on a dry block for 5 minutes) and freeze-thaw (3 cycles) were subjected to DNA extraction following treatment with PMA or the DMSO (20%) diluent. Samples

**Table 1. Demographic information of the dogs that provided stool samples.**

| ID | Age (Y)[*] | Sex[**] | Neuter status | BW (kg)[***] | Breed |
|---|---|---|---|---|---|
| 1 | 4.5 | F | N | 13.0 | Beagle |
| 2 | 5 | F | N | 34.1 | Golden Retriever |
| 3 | 4 | M | N | 3.4 | MBD |
| 4 | 2 | F | N | 12.7 | MBD |
| 5 | 7 | M | N | 27.7 | Siberian Husky |
| 6 | 6 | F | N | 20.5 | MBD |
| 7 | 6 | F | N | 9.5 | French Bulldog |
| 8 | 6 | M | I | 23.4 | French Brittany |
| 9 | 4.5 | F | N | 23.6 | MBD |
| 10 | 3 | M | N | 28.6 | Labrador Retriever |
| 11 | 4 | M | N | 13.6 | Pembroke Corgi |
| 12 | 4 | M | N | 29.8 | Goldendoodle |
| 13 | 3 | M | N | 40.9 | Labrador Retriever |
| 14 | 8 | M | N | 8.0 | Yorkie-poo |
| 15 | 4 | M | N | 28.2 | Labrador Retriever |
| 16 | 5 | F | N | 31.8 | Greyhound |
| 17 | 4 | M | N | 2.3 | Havanese |
| 18 | 2 | F | N | 11.4 | Pembroke Corgi |
| 19 | 3 | F | N | 5.1 | MBD |
| 20 | 2.6 | M | I | 56.8 | Rottweiler |

**F**, female; **M**, male; **I**, intact; **N**, neutered; **MBD**, mixed breed dog.

*Mean (±SD) age (y) = 4.4 ± 1.6

**Male dogs did not significantly differ in number from female dogs (p = 0.655)

***Mean (±SD) bodyweight (kg) = 21.2 ± 14

of glycerol, PBS, and nuclease-free water were submitted for sequencing to detect reagent contamination (negative controls).

## PMA treatment and DNA extraction

1 mg of PMA stock (Biotium Inc., Hayward, CA) was diluted in 1 mL 20% DMSO in sterile, DNA/RNA/nuclease-free water. 1 mL of each aliquot was diluted in 1×PBS for a final volume of 5 mL and mixed with 110 μL of diluted PMA (43μM). To prevent light contamination, the samples were incubated with gentle tilting (6 rpm) at room temperature for 30 min in a styrofoam cooler lined with aluminum foil. The samples were then incubated for 20 minutes under full light from a LED aquarium light source affixed to the open cooler top. The samples were centrifuged at 600× $g$ for 10 minutes at 4˚C, and the supernatant was discarded. The pellets were washed with 1 mL 1× PBS, centrifuged at 600× $g$ for 10 minutes at 4˚C, and the supernatant was discarded. Bulk genomic DNA was purified from 200 mg of the fecal pellet using the QIAGEN Power Fecal Pro DNA kit (Qiagen Inc., Hilden, Germany) per the manufacturer's protocol with an additional incubation step at 65˚C for 5 minutes before tubes were vortexed for 10 minutes. Genomic DNA quantity and quality were assessed via fluorometry (Qubit fluorometer; ThermoFisher, CA) and electrophoresis in a 1% agarose gel, respectively. DNA from the baseline, 3-month, 6-month, 9-month, and 12-month samples were stored at -80˚C.

## Power-sample size analysis

Due to the high volume and complexity of data generated in high-throughput sequencing studies (millions of sequencing reads), there are no commonly accepted a priori methods of sample size estimation and existing methods do not account for repeated measures [21, 22]. The sample size for this study is based on previous investigations of FMT (human and mouse origin) in which repeated measures were implemented to compare bacterial viability and beta diversity by conventional laboratory culture methods and 16S rRNA-based bioinformatic methods [8, 9, 23]. Using analytic methods that account for repeated measures, these studies generated statistically significant and biologically relevant results using 4, 5, and 21 fecal samples per group [8, 9, 23]. In order to improve the discriminatory power in this study, we enrolled a total of 20 dogs.

## Long-read 16S-rRNA amplicon sequencing

Library construction and sequencing with the PacBio Sequel II was performed at the Roy J. Carver Biotechnology Center, University of Illinois at Urbana-Champaign. The 16S rRNA gene amplicons were generated with the Shoreline Complete ID kit (Shoreline Biome), which amplifies a 2,500 bp fragment including the full 16S, the intergenic sequence (ITS) and a portion of the 23S rRNA gene. The kit contains a patented mix of forward and reverse primers. The consensus sequence of the primers is 5′-AGRRTTYGATYHTDGYTYAG-3′ (forward) and 5′-AGTACYRHRARGGAANGR-3′ (reverse). Individually barcoded amplicons were combined into two pools. Each pool was converted to a barcoded Pacbio library with the SMRTBell Express Template Prep kit 2.0 (Pacific Biosciences). The libraries were sequenced on two SMRTcell 8M in the PacBio Sequel II using the CCS sequencing mode and 15hs movie times.

The resulting FASTQ reads were processed in two steps. Initial FASTQ sequence data was demultiplexed per sample using SBAnalyzer v3.1 from Shoreline Biosciences (now Intus Biosciences, https://intusbio.com/). Data were processed to retain the primer sequences in the demultiplexed reads so the next step could properly reorient the final sequence data. Demultiplex read data were further processed using a Nextflow-based workflow, TADA [24]. TADA automates using DADA2 v1.22 for trimming and denoising reads based on the protocols used

for PacBio data to generate amplicon sequence variants (ASVs) [25]. The specific run here used github checkout 18bd4fab which includes support for processing Shoreline data. Taxonomic assignment utilized two databases. First, we used the DADA2 implementation of the RDP classifier [26] to classify reads using the SILVA 138.1 release, with a database formatted for PacBio HiFi read data (https://zenodo.org/record/4587955). Additionally, the command-line version of SBAnalyzer was used to identify taxonomic matches at the strain level using Shoreline's Athena v2.2 database, which includes 16S-ITS-23S sequence data derived from publicly available microbial genomes [27]. The Shoreline Athena databases is thought to be more specific at the level of species and sub-species/strain. Owing to differences in the taxonomic annotations between the SILVA and the Shoreline Athena databases, we presented the taxonomic annotations from both databases. The long-read 16S-rRNA amplicon sequences are deposited in the National Center for Biotechnology Information (NCBI) BioProject repository (https://www.ncbi.nlm.nih.gov/bioproject/PRJNA1001120) under the accession number PRJNA1001120.

## Statistical analysis

Statistical analysis of sample DNA concentrations were performed using SAS® OnDemand for Academics (SAS Institute Inc., Cary, NC, USA). Sample DNA concentration was examined for normal distribution by inspection of Q-Q plots and histogram, and the Shapiro-Wilk test and was then log-transformed to assume a log-normal distribution. Analysis of variance for the log-transformed DNA concentration and Δ log-transformed DNA concentration from baseline (baseline *log* DNA—timepoint *log* DNA) were performed with the MIXED procedure. The model included the fixed effects of timepoint, storage temperature, timepoint × storage temperature interaction, and the random effect of the dog to account for the repeated measurements on fecal DNA samples from the same dog. The effects of timepoint, storage temperature, and timepoint × storage temperature interaction on the mean of dependent variables were analyzed by the Fisher Least Significant Difference test with Tukey posthoc correction as implemented in the LSMEANS option. The statistical significance level was set on $p \leq 0.05$.

Statistical analysis of 16S-rRNA amplicon sequencing was performed in the R Language for Statistical Computing (The R Foundation; v4.2.1). Data and R code sufficient to replicate this analysis are in our github repository (https://github.com/pcbarko/K9 FMT Storage Stability). To detect and remove any contaminating ASVs identified in the negative control samples, the "decontam" script was used as previously reported [28]. Alpha diversity was measured using the Shannon diversity index (SDI) on the unfiltered count matrix, implemented in the "phyloseq" package [29]. Differences in SDI associated with storage time and temperature was compared using linear mixed-effects models. ASVs with an unknown phylum (n = 346) were removed to exclude ASVs with unknown taxonomy. To exclude singletons and other features present in a only small numbers of samples, the data were filtered to exclude ASVs with a total abundance < 1 and those present in fewer than 5% of samples. The Bray-Curtis dissimilarity index was calculated using the "phyloseq" package on filtered count matrices after normalization to relative abundance. Differences in beta diversity were compared among groups using permutational multivariate analysis of variance (PERMANOVA) in the "vegan" package [30]. Non-metric multidimensional scaling (NMDS) was used to visualize the Bray-Curtis dissimilarity matrix. For differential abundance analysis the filtered feature count was agglomerated to the taxonomic level of species, normalized to relative abundance by total sum scaling, and transformed by arcsine square root transformation. To detect differentially abundant features due to storage temperature and time, microbiome multivariable associations with linear models

(MaAsLin2) was implemented using the "maaslin2" package [31]. In the MaAsLin2 model, storage time and temperature were fixed effects and the individual dog was used as a random effect.

To determine whether fecal microbiomes from different dogs were differentially stable during storage, we assessed multivariate homogeneity of beta dispersions on all samples (fresh and stored) implemented in the "vegan" package. To quantify dispersion (variance) of fecal microbiome profiles the mean distance of samples to the centroid for each individual dog was calculated in multivariate space generated by the Bray-Curtis dissimilarity matrix. ANOVA with post-hoc Tukey's HSD were used to detect statistically significant differences in dispersion among dogs. Using the beta dispersion statistics, each dog was classified as either "low-dispersion" or "high-dispersion" based on comparing the mean distances to centroid for each dog with the overall mean distance to centroid for the entire cohort. Dogs with a mean distance to centroid greater than the overall mean were classified as "high-dispersion" and those with median distance to centroid lesser than the overall mean were classified as "low-dispersion." To determine if alpha diversity in the baseline samples was associated with variability of microbiome profiles during storage, we compared alpha diversity indices (Shannon index, observed species) from the baseline samples among the high and low-dispersion dogs using Wilcoxon rank sum tests. To determine if microbiome composition in the baseline samples was associated with variability of microbiome profiles during storage, differential abundance of ASVs between the low and high-dispersion groups was assessed using MaAsLin2. To control for false discovery due to multiple comparisons, we calculated an estimate of the false discovery rate (FDR; q-value) as previously described [32]. For features to be considered significant we used a q-value threshold of q $\leq$ 0.05.

## Results

### Fecal DNA concentrations

Marginal mean (±SE) *log* fecal DNA (ng/μL) across all timepoints in -20˚C storage (2.8 ± 0.2) was significantly lower than in -80˚C storage (3.4 ± 0.2; p < 0.001).

Marginal mean (±SE) *log* fecal DNA (ng/μL) in -20˚C storage at baseline (3.8 ± 0.3) was significantly higher than at 3-month (1.9 ± 0.3; p < 0.001), 6-month (3.1 ± 0.3; p = 0.012), 9-month (2.6 ± 0.3; p < 0.001), and 12-month (2.6 ± 0.3; p < 0.001).

Marginal mean (±SE) *log* fecal DNA (ng/μL) in -80˚C storage at baseline (3.8 ± 0.3) was significantly higher than at 3-month (2.8 ± 0.3; p < 0.001) and 12-month (3.0 ± 0.3; p = 0.002) but did not differ from the 6-month (3.5 ± 0.3; p = 0.921) and 9-month (3.6 ± 0.3; p = 0.994).

There were significant differences in Δ marginal mean (±SE) *log* fecal DNA from baseline (i.e., baseline *log* DNA—timepoint *log* DNA) between storage conditions (p < 0.001), within the same storage condition across timepoints (p < 0.001), and for the interaction between storage conditions × timepoints (Fig 1A; p = 0.012). Mean (±SD) fecal DNA concentrations of the 20 stool samples included in this study across all timepoints and storage conditions are reported in Fig 1B.

### 16S-rRNA amplicon sequencing and microbiome analysis

6,004,861 sequences were generated and binned into 3,928 amplicon sequence variants (ASV). Three hundred forty-six ASVs with unidentified phyla were removed. Only 158 sequences were detected in the negative control samples (glycerol, PBS, nuclease-free water), indicating a low level of reagent contamination. Two ASVs present in the negative control samples, identified as *Catenibacterium mitsuokai*, were detected as contaminants and removed from the feature table prior to analysis. Though the two ASVs assigned to *Catenibacterium mitsuokai* were detected as potential contaminants in the negative control samples and removed prior to

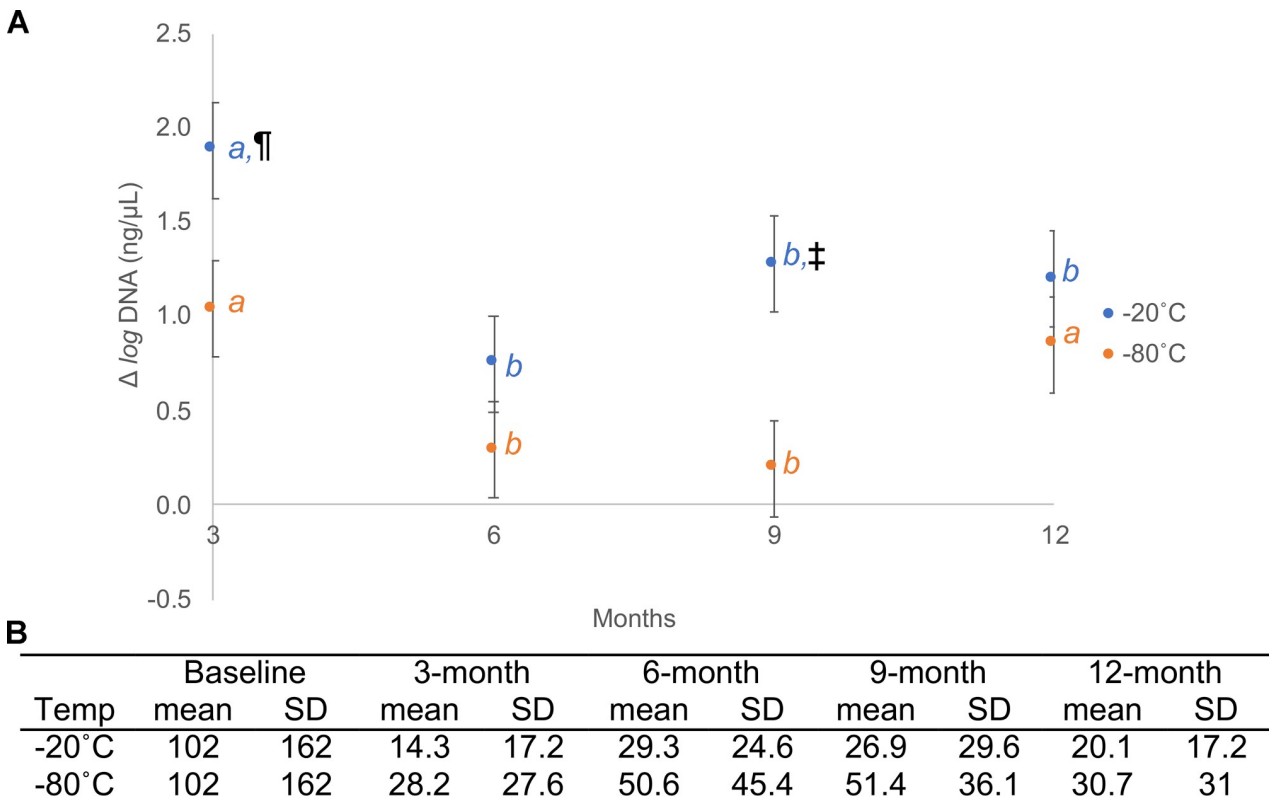

**Fig 1. Fecal DNA concentrations (ng/μL) across time and storage conditions. 1A**. Δ Mean (±SE) *log* fecal DNA concentration (ng/μL) from baseline (i.e., baseline *log* DNA—timepoint *log* DNA) in -20˚C and -80˚C storage. Different italicized lower-case letters represent a significant difference (p < 0.05) between timepoints within the same storage temperature. ¶ and ‡ represent a significant difference (p < 0.05) between storage temperatures within the same timepoint. **1B**. Mean (±SD) fecal DNA concentration (ng/μL) of the 20 stools samples included in this study.

Table B:

| Temp | Baseline | | 3-month | | 6-month | | 9-month | | 12-month | |
|---|---|---|---|---|---|---|---|---|---|---|
| | mean | SD | mean | SD | mean | SD | mean | SD | mean | SD |
| -20˚C | 102 | 162 | 14.3 | 17.2 | 29.3 | 24.6 | 26.9 | 29.6 | 20.1 | 17.2 |
| -80˚C | 102 | 162 | 28.2 | 27.6 | 50.6 | 45.4 | 51.4 | 36.1 | 30.7 | 31 |

analysis, *Catenibacterium mitsuokai* was also detected as a differentially abundant bacteria. This is possible because there were originally 20 ASV assigned to *Catenibacterium mitsuokai*. After removing the two contaminating ASVs, 18 ASVs with the same taxonomic assignment remained. These were agglomerated to the level of species and included in the differential abundance analysis. After filtering to remove features with zero counts that were not present in at least 5% of samples there remained 339 ASVs. The filtered and agglomerated count matrix contained features from 40 species from 22 distinct genera. For the PMA/lysis control samples, NMDS of the Bray-Curtis dissimilarity matrix did not reveal separation of samples exposed to lysis conditions with or without PMA, but larger proportions of Fusobacteria and Bacteroidota were present in samples treated with PMA. As no biologic replicates were available, inferential statistics were not performed on the lysis control samples.

The Shannon alpha diversity indices of the samples stored at -80˚C were significantly higher than in -20˚C storage for each of the study timepoints (Fig 2 and Table 2; p < 0.05). There were no statistically significant differences in Shannon alpha diversity indices among any timepoints (including baseline) for samples stored at -80˚C. The Shannon alpha diversity index was significantly higher at baseline and 3-months compared to the 6-month, 9-month, and 12-month sampled stored in -20˚C (p < 0.05).

Visualization of the Bray-Curtis dissimilarity (beta diversity) matrix using NMDS revealed separation of samples according to the individual dog of origin. PERMANOVA computed on Bray-Curtis dissimilarity matrices revealed significant differences among samples collected

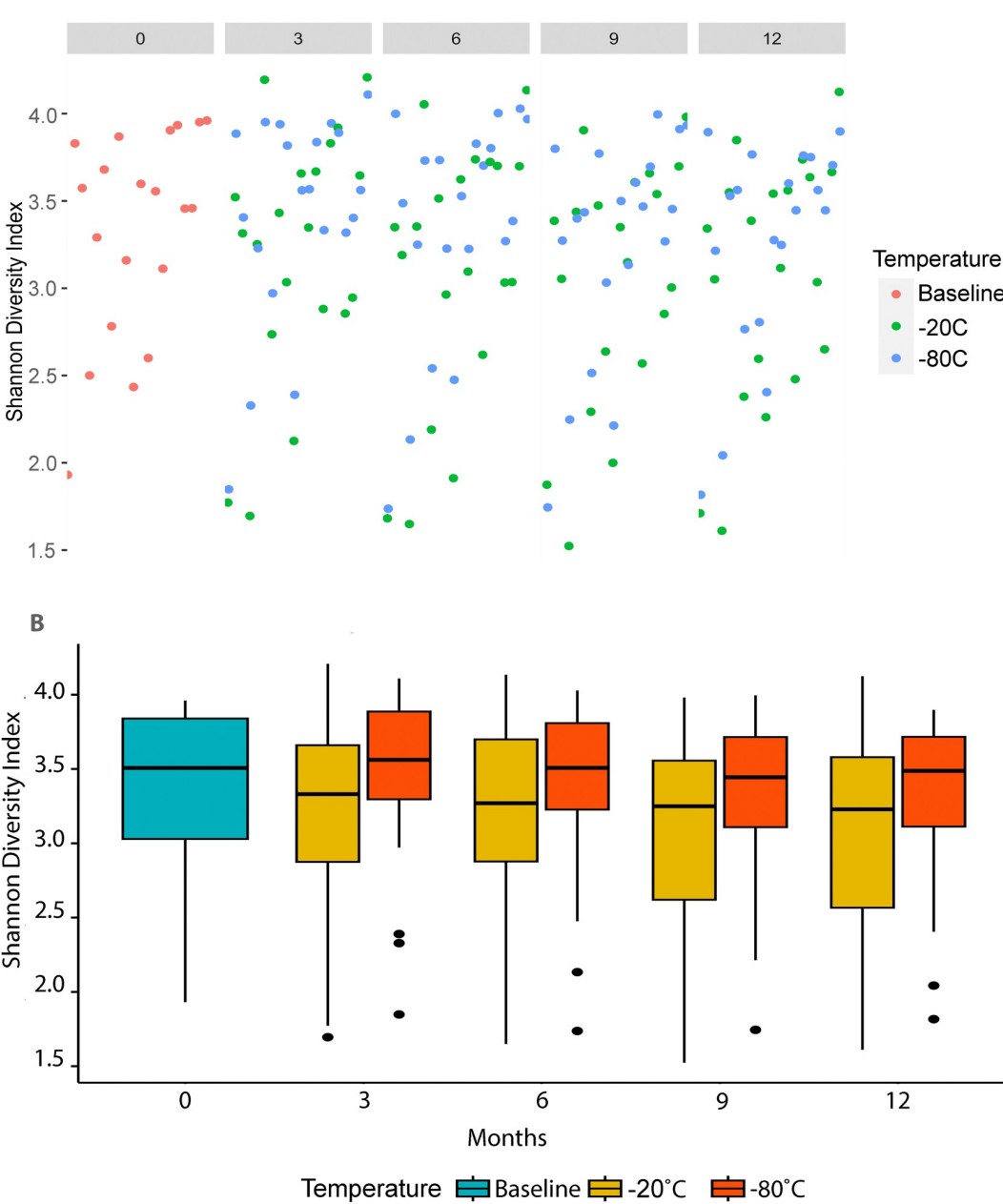

**Fig 2. Shannon alpha diversity index of 20 canine stools samples in -20˚C and -80˚C storage for 12 months. 2A.** Scatter plot of individual samples. **2B.** Boxplots. The edges of the box represent the 25th and 75th percentiles. The whiskers represent the maximum and minimum values below and above the upper (75th percentile + IQR) and lower (and 25th percentile–IQR) fences, respectively.

from individual dogs (Fig 3; PERMANOVA F-statistic = 97.0, $p < 0.001$). There was no apparent separation of samples due to storage temperature or time in the NMDS plots (Fig 3), however, significant effects of storage temperature (PERMANOVA F-statistic = 0.87, $p < 0.001$)

**Table 2. Marginal mean (±SE) of the Shannon diversity index of 20 canine stools samples in -20˚C and -80˚C storage for up to 12 months.**

| Timepoint | Storage (˚C) | Marginal mean | SE | P value[*] |
|---|---|---|---|---|
| Baseline | NA | 3.33 | 0.15 | NA |
| 3-month | -20 | 3.20 | 0.15 | 0.035 |
| | -80 | 3.41 | 0.15 | |
| 6-month | -20 | 3.11 | 0.15 | 0.009 |
| | -80 | 3.35 | 0.15 | |
| 9-month | -20 | 3.05 | 0.15 | 0.025 |
| | -80 | 3.27 | 0.15 | |
| 12-month | -20 | 3.06 | 0.15 | 0.039 |
| | -80 | 3.27 | 0.15 | |

[*]Represents the difference between groups within timepoint.

**SE**, standard error; **NA**, not applicable.

and time (PERMANOVA F-statistic = 0.17, p < 0.001) were observed using PERMANOVA to compare Bray-Curtis dissimilarity matrices among the different storage conditions (Fig 3).

The relative abundance of 15 species features varied significantly (q ≤ 0.05) in association with storage temperature when adjusting for the storage time covariate (Table 3 and S1 Fig). Relative abundances of *Fusobacterium gastrosuis*, *Fusobacterium perfoetens*, *Fusobacterium necrogenes*, *Megamonas funiformis*, *Alloprevotella rava*, and an unclassified species of Fusobacterium were significantly decreased, whereas *Blautia glucerasea*, an unclassified species of *Lactobacillus*, *Romboutsia lituseburensis*, *Streptococcus pasteuri*, *Catenibacterium mitsuokai*, *Allobaculum stercoricanis* and *Turicibacter sanguinis* were significantly increased in samples stored at -20˚C compared with baseline feces. Relative abundances of *Fusobacterium gastrosuis*, *Fusobacterium perfoetens*, *Fusobacterium necrogenes*, *Megamonas funiformis*, *Alloprevotella rava*, and an unclassified species of *Fusobacterium* were significantly decreased, and *Blautia glucerasea*, an unclassified species of *Lactobacillus*, an unclassified species of *Erysipelatoclostridium*, *Romboutsia lituseburensis*, *Streptococcus pasteuri*, *Catenibacterium mitsuokai*, and *Turicibacter sanguinis* were significantly increased, in samples stored at -20˚C compared with -80˚C.

Adjusting for the storage temperature covariate the relative abundances of 13 features were significantly (q ≤ 0.05) variable in association with time (Table 4 and S1 Fig). Compared with baseline samples not exposed to storage, the relative abundances of *Megamonas funiformis* was significantly lower at 3 months, 6 months, 9 months, and 12 months. Compared with storage for 3 months the relative abundances of *Alloprevotella rava* was lower at 6 months, 9 month, and 12 months; *Bacteroides massiliensis* was lower at 6 months; *Allobaculum stercoricanis* was lower at 9 months; and *Fusobacterium perfoetens* was lower at 12 months. Compared with storage for 6 and 9 months the relative abundances of *Asaccharospora irregularis* and an unclassified species of *Asaccharospora* were lower at 12 months. Compared with baseline samples not exposed to storage the relative abundances of *Collinsella intestinalis* were higher at 3 months and 12 months; *Blautia glucerasea* and *Catenibacterium mitsuokai* were higher at 6 months; *Holdemanella biformis* and *Turicibacter sanguinis* were increased at 12 months. Compared with samples stored at 6 and 9 months the relative abundance of *Turicibacter sanguinis* and *Collinsella intestinalis* were higher after 12 months of storage.

Beta dispersion varied significantly among individual dogs and there were numerous statistically significant differences between individual dogs in the post-hoc tests (Fig 4 and S1 Table; p < 0.001). 9 dogs were classified into the high-dispersion group, whereas the remaining 11

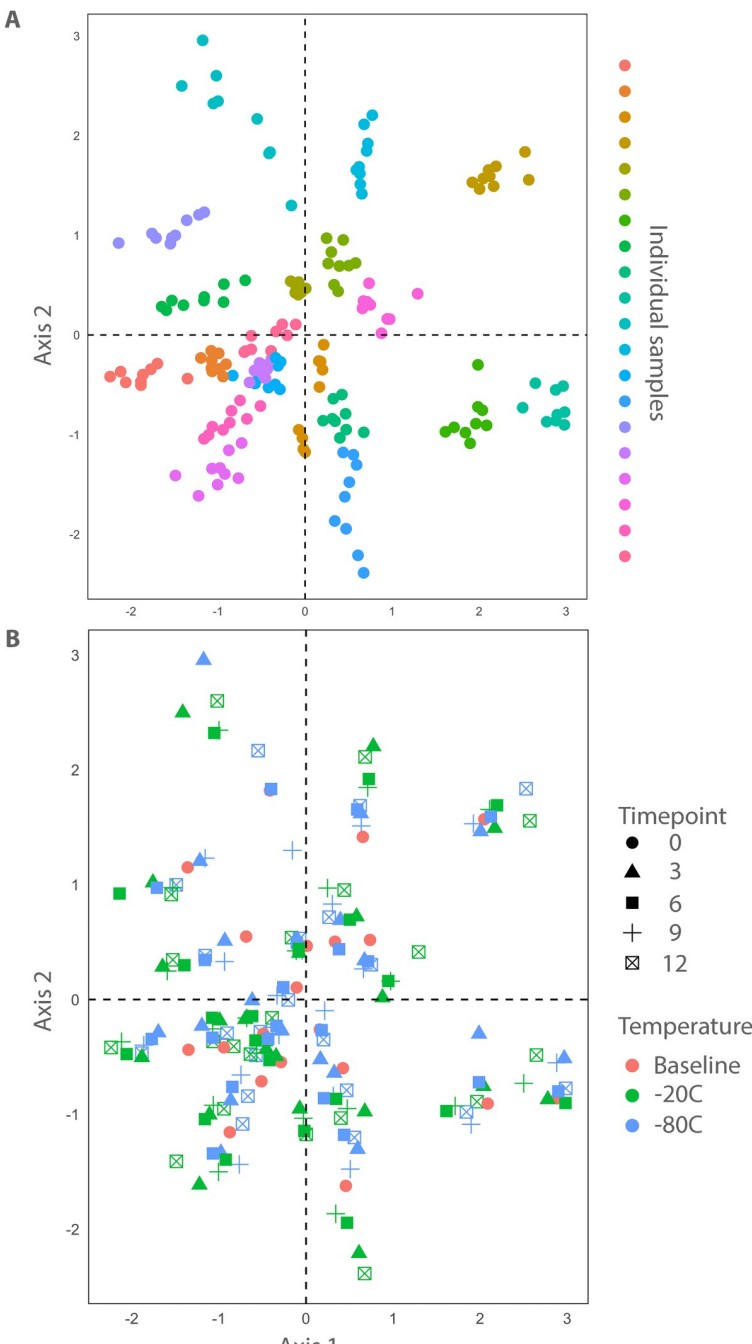

**Fig 3. Bray Curtis distances visualized with nonmetric multidimensional scaling (NMDS) plots of 20 stool samples in -20°C and -80°C storage for 12 months. 3A**. Individual stool samples. **3B**. Presentation by storage condition and timepoint.

dogs were classified into the low-dispersion group. Shannon diversity indices were significantly higher (p = 0.02) in the high-dispersion compared with the low-dispersion group, but were no significant differences in the observed species index (p = 0.21) between the high-dispersion or low-dispersion groups. An unclassified species of *Fusobacterium* was more abundant (q ≤ 0.033) in the fresh feces of dogs with high compared with low beta dispersion.

**Table 3. Features that varied significantly with temperature.**

| Taxa (Silva database) | Taxa (Athena database) | Contrast | Model Coefficient | p-value | q-value |
|---|---|---|---|---|---|
| *Fusobacterium gastrosuis* | *bacterium unclassified* | Baseline vs -20˚C | -0.12 | <0.001 | <0.001 |
| *Fusobacterium perfoetens* | *Bacillus unclassified* | Baseline vs -20˚C | -0.11 | <0.001 | <0.001 |
| *Fusobacterium necrogenes* | *Fusobacterium mortiferum ATCC 9817* | Baseline vs -20˚C | -0.06 | <0.001 | <0.001 |
| *Megamonas funiformis* | *Megamonas unclassified* | Baseline vs -20˚C | -0.05 | 0.006 | 0.032 |
| *Alloprevotella rava* | *Prevotellaceae unclassified* | Baseline vs -20˚C | -0.03 | <0.001 | <0.001 |
| *Fusobacterium unclassified* | *Fusobacterium unclassified* | Baseline vs -20˚C | -0.02 | <0.001 | <0.001 |
| *Blautia glu35cerasea* | *Blautia unclassified* | Baseline vs -20˚C | 0.01 | 0.004 | 0.024 |
| *Unclassified Unclassified* | *Lactobacillus unclassified* | Baseline vs -20˚C | 0.02 | 0.003 | 0.022 |
| *Erysipelatoclostridium Unclassified* | *Erysipelatoclostridium unclassified* | Baseline vs -20˚C | 0.02 | <0.001 | <0.001 |
| *Romboutsia lituseburensis* | *Intestinibacter unclassified* | Baseline vs -20˚C | 0.03 | <0.001 | <0.001 |
| *Streptococcus pasteuri* | *Intestinibacter unclassified* | Baseline vs -20˚C | 0.03 | <0.001 | 0.002 |
| *Catenibacterium mitsuokai* | *Catenibacterium mitsuokai DSM 15897* | Baseline vs -20˚C | 0.05 | <0.001 | <0.001 |
| *Turicibacter sanguinis* | *Turicibacter sp. H121 unclassified* | Baseline vs -20˚C | 0.07 | <0.001 | <0.001 |
| *Allobaculum stercoricanis* | *Allobaculum stercoricanis DSM 13633* | Baseline vs -20˚C | 0.01 | 0.009 | 0.049 |
| *Fusobacterium gastrosuis* | *Fusobacterium unclassified* | -80˚C vs -20˚C | -0.12 | <0.001 | <0.001 |
| *Fusobacterium perfoetens* | *Bacillus unclassified* | -80˚C vs -20˚C | -0.11 | <0.001 | <0.001 |
| *Fusobacterium necrogenes* | *Bacillus unclassified* | -80˚C vs -20˚C | -0.06 | <0.001 | <0.001 |
| *Megamonas funiformis* | *Megamonas unclassified* | -80˚C vs -20˚C | -0.05 | <0.001 | 0.005 |
| *Alloprevotella rava* | *Prevotellaceae unclassified* | -80˚C vs -20˚C | -0.03 | <0.001 | <0.001 |
| *Fusobacterium unclassified* | *Fusobacterium unclassified* | -80˚C vs -20˚C | -0.02 | <0.001 | <0.001 |
| *Blautia glu55cerasea* | *Blautia unclassified* | -80˚C vs -20˚C | 0.01 | 0.002 | 0.012 |
| *Unclassified Unclassified* | *Lactobacillus unclassified* | -80˚C vs -20˚C | 0.02 | 0.002 | 0.012 |
| *Erysipelatoclostridium unclassified* | *Erysipelatoclostridium unclassified* | -80˚C vs -20˚C | 0.02 | <0.001 | <0.001 |
| *Romboutsia lituseburensis* | *Intestinibacter unclassified* | -80˚Cvs -20˚C | 0.02 | <0.001 | <0.001 |
| *Streptococcus pasteuri* | *Streptococcus unclassified* | -80˚C vs -20˚C | 0.03 | <0.001 | <0.001 |
| *Catenibacterium mitsuokai* | *Catenibacterium mitsuokai DSM 15897* | -80˚C vs -20˚C | 0.05 | <0.001 | <0.001 |
| *Turicibacter sanguinis* | *Turicibacter sp. H121 unclassified* | -80˚C vs -20˚C | 0.07 | <0.001 | <0.001 |
| *Allobaculum stercoricanis* | *Allobaculum stercoricanis DSM 13633* | -80˚C vs -20˚C | 0.01 | 0.012 | 0.050 |

## Discussion

Acute and chronic enteropathies are significant causes of presentation for primary veterinary care worldwide [33–35] with a reported prevalence of 5%-7% for acute [36–39], and 10%-17% for chronic enteropathies [40–42]. In both acute and chronic enteropathies, there are significant alterations in the composition of gut microbiome communities (i.e., dysbiosis) [43–56]. In dysbiosis, functional alterations in the gut's microbial transcriptome, proteome, and metabolome could damage the host gastrointestinal tract [57]. Evidence from humans, canines, and rodents suggest that dysbiosis can be ameliorated by dietary interventions [58, 59], administration of prebiotics [59, 60], probiotics [59, 61], or by fecal microbiota transplantation [12–20], reserving antibiotic therapy for selected specific indications.

Fecal microbial transplantation is an effective, FDA-approved treatment for recurrent *Clostridioides difficile*-associated colitis infections in humans, a disorder characterized by profound disruption in the community composition of commensal bacteria and decreased microbiota diversity [62–64]. While human fecal donors are heavily screened for a large panel of infectious pathogens to prevent their transmission to FMT recipients, they are not routinely screened for fecal microbiome diversity or composition [65]. Similarly, there has been no systematic attempts to assess donor dogs' microbial diversity before the administration of FMT in dogs

**Table 4. Features that varied significantly with storage time.**

| Taxa (Silva database) | Taxa (Athena database) | Contrast | Model Coefficient | p-value | q-value |
|---|---|---|---|---|---|
| *Megamonas funiformis* | *Megamonas unclassified* | Baseline vs 3 months | -0.13 | <0.001 | <0.001 |
| *Collinsella intestinalis* | *Collinsella intestinalis DSM 13280* | Baseline vs 3 months | 0.05 | <0.001 | 0.006 |
| *Megamonas funiformis* | *Collinsella intestinalis DSM 13280* | Baseline vs 3 months | -0.11 | <0.001 | 0.002 |
| *Blautia glucerasea* | *Blautia unclassified* | Baseline vs 6 months | 0.01 | 0.006 | 0.031 |
| *Catenibacterium mitsuokai* | *Catenibacterium mitsuokai DSM 15897* | Baseline vs 6 months | 0.05 | 0.003 | 0.020 |
| *Megamonas funiformis* | *Megamonas unclassified* | Baseline vs 9 months | -0.09 | 0.005 | 0.029 |
| *Megamonas funiformis* | *Megamonas unclassified* | Baseline vs 12 months | -0.12 | <0.001 | 0.001 |
| *Holdemanella biformis* | *Faecalitalea unclassified* | Baseline vs 12 months | 0.04 | 0.002 | 0.015 |
| *Turicibacter sanguinis* | *Turicibacter sp. H121 unclassified* | Baseline vs 12 months | 0.06 | 0.001 | 0.010 |
| *Collinsella intestinalis* | *Collinsella intestinalis DSM 13280* | Baseline vs 12 months | 0.08 | <0.001 | <0.001 |
| *Alloprevotella rava* | *Prevotellaceae unclassified* | 3 months vs 6 months | -0.05 | <0.001 | <0.001 |
| *Bacteroides massiliensis* | *Bacteroides unclassified* | 3 months vs 6 months | -0.02 | 0.006 | 0.029 |
| *Erysipelatoclostridium unclassified* | *Erysipelatoclostridium unclassified* | 3 months vs 6 months | 0.03 | <0.001 | <0.001 |
| *Asaccharospora unclassified* | *Clostridioides hiranonis unclassified* | 3 months vs 6 months | 0.04 | <0.001 | 0.002 |
| *Asaccharospora irregularis* | *Clostridioides hiranonis unclassified* | 3 months vs 6 months | 0.05 | <0.001 | 0.005 |
| *Alloprevotella rava* | *Prevotellaceae unclassified* | 3 months vs 9 months | -0.03 | 0.010 | 0.042 |
| *Allobaculum stercoricanis* | *Allobaculum stercoricanis DSM 13633* | 3 months vs 9 months | -0.02 | 0.002 | 0.012 |
| *Erysipelatoclostridium unclassified* | *Erysipelatoclostridium unclassified* | 3 months vs 9 months | 0.02 | <0.001 | 0.006 |
| *Asaccharospora unclassified* | *Clostridioides hiranonis unclassified* | 3 months vs 9 months | 0.03 | <0.001 | 0.005 |
| *Asaccharospora irregularis* | *Clostridioides hiranonis unclassified* | 3 months vs 9 months | 0.06 | <0.001 | <0.001 |
| *Fusobacterium perfoetens* | *Erysipelatoclostridium unclassified* | 3 months vs 12 months | -0.05 | 0.006 | 0.028 |
| *Alloprevotella rava* | *Erysipelatoclostridium unclassified* | 3 months vs 12 months | -0.03 | 0.003 | 0.018 |
| *Erysipelatoclostridium unclassified* | *Erysipelatoclostridium unclassified* | 3 months vs 12 months | 0.02 | 0.002 | 0.010 |
| *Asaccharospora irregularis* | *Clostridioides hiranonis unclassified* | 6 months vs 12 months | -0.05 | <0.001 | 0.002 |
| *Asaccharospora unclassified* | *Clostridioides hiranonis unclassified* | 6 months vs 12 months | -0.03 | 0.002 | 0.011 |
| *Turicibacter sanguinis* | *Turicibacter sp. H121 unclassified* | 6 months vs 12 months | 0.04 | 0.003 | 0.018 |
| *Collinsella intestinalis* | *Collinsella intestinalis DSM 13280* | 6 months vs 12 months | 0.04 | <0.001 | 0.004 |
| *Asaccharospora irregularis* | *Clostridioides hiranonis unclassified* | 9 months vs 12 months | -0.06 | <0.001 | <0.001 |
| *Asaccharospora unclassified* | *Clostridioides hiranonis unclassified* | 9 months vs 12 months | -0.03 | 0.005 | 0.031 |
| *Collinsella intestinalis* | *Collinsella intestinalis DSM 13280* | 9 months vs 12 months | 0.04 | 0.001 | 0.012 |
| *Turicibacter sanguinis* | *Turicibacter sp. H121 unclassified* | 9 months vs 12 months | 0.04 | 0.004 | 0.025 |

[12–18, 20]. Studies of FMT in humans indicate that selection of an appropriate stool donor, impacts responses to FMT [66, 67], and that fecal microbiota diversity is a reliable predictor of FMT efficacy [67, 68].

In humans, it has been shown that FMT of fresh-frozen feces has similar efficacy as fresh feces in the resolution of recurrent *Clostridioides difficile*-associated colitis [7–11]. To facilitate large-scale implementation of FMT in patients with recurrent *Clostridioides difficile*-associated colitis, fresh-frozen stool banks have been established. The advantages of stored feces include increased availability and accessibility of high-quality donor fecal material. Yet, to our knowledge, the stability of frozen canine feces, with respect to fecal microbiome composition and diversity, has not been explored. Likewise, criteria to identify ideal donor dogs have not been established.

We demonstrated that fresh-frozen stool from healthy dogs in 20% glycerol can be stored for up to 12 months at -80˚C with minimal change in microbial composition and diversity. Further, there were no differences in alpha diversity in storage at -80˚C for up to 12 months, whereas storage at -20˚C was associated with lower alpha diversity. We also demonstrated that

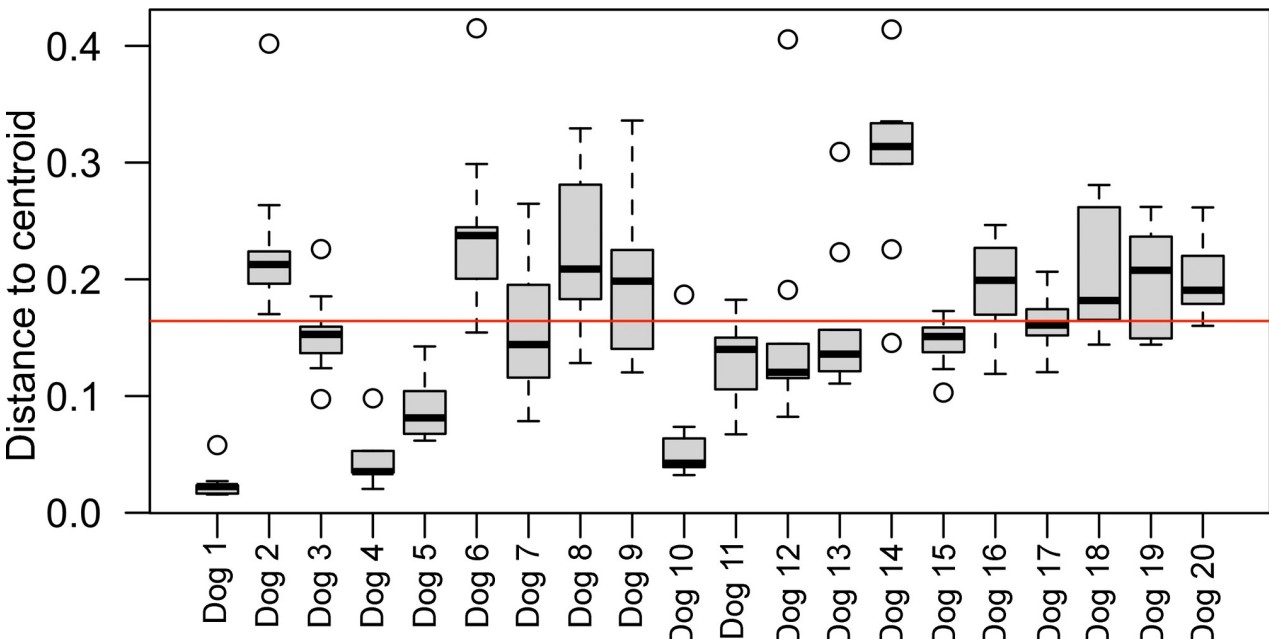

**Fig 4. Multivariate homogeneity of beta dispersions.** Beta dispersions were calculated from Bray-Curtis dissimilarity matrix to determine if variance in fecal microbiomes among storage conditions differed among individual dogs. The boxplots show beta dispersion in each dog with the upper and lower boundaries of the box represent the 25th and 75th percentiles and the horizontal line represents the median. The whiskers represent the maximum and minimum values below and above the upper (75th percentile + IQR) and lower (and 25th percentile–IQR) fences, respectively. The mean beta dispersion is represented by the horizontal red line.

there were significant differences in beta diversity associated with both storage time and temperature (Fig 3 and Tables 3, 4 and S1 Fig). Presumably, the therapeutic benefit of FMT is mostly mediated by the transplantation of viable microbes that can affect gut microbial community composition and diversity in the recipient. However, the assessment of bacterial viability is complex, and complicated by the fact that most of the gut microbiome cannot be cultured under standard laboratory methods. As the typical amplicon and metagenomic sequencing studies utilize bulk DNA extracts that contain genetic material from both bacteria with intact and damaged cell membranes, we chose to tackle this problem using a well-established method employing propidium monoazide. Propidium monoazide is a photoreactive dye that preferentially binds double-stranded DNA and under light activation renders it inert as a substrate for downstream sequencing. PMA cannot penetrate bacteria with intact cell membranes but can bind free DNA if the bacterial cell membranes are damaged. This assures that in PMA-treated samples only genetic material from bacteria with intact cell membranes can be quantified by downstream sequencing. As all samples were treated with PMA, we demonstrated that the gut microbial communities in our samples were derived from bacteria with intact cell membranes and were preserved for up to 12 months in both storage temperatures.

The major limitation of the approach that we took is that we could not determine if storage duration or conditions had resulted in any storage-related epigenetic or other functional changes that would make cryopreserved bacteria lose their ability to maintain their normal biological activities post-thawing in the recipient animal's intestinal tract. Such an approach should be considered as the next layer of investigation in future studies on cryopreserved stool intended for FMT. Furthermore, owing to evidence that FMT of cellular extracts derived from dead bacteria can have a desirable biological effect on FMT recipient [5], the application of PMA on our samples did not account for any potential biological effects derived from the

portion of donor fecal material that may have contained dead bacteria's cellular extracts, had these cryopreserved stool samples been used for FMT. An additional limitation inherent to our fecal processing protocol involved several centrifugation steps that may have resulted in loss of bacteria that resided in the supernatant or alternatively resulted in loss of strict anaerobes owing to the additional contact time with air. We neither determined the DNA content nor sequenced the supernatant that we discarded so we do not know if and how many bacteria were lost, and we suspect that it could have some impact on our results. Lastly, we did not record the diet and geographical locations where the dogs resided. Both diet and soil bacteria from different geographic locations could have impacted the gut microbial community structure of the dogs. Including these factors in the multivariate analysis potentially could have revealed important information about their effect on the stability of cryopreserved stool.

We demonstrated that the relative abundances of certain bacteria were affected by storage temperature (Table 3) and duration (Table 4). For example, the relative abundances of *Fusobacteria* and *Megamonas* were significantly higher in fresh samples and those stored at-80˚C, whereas short-chain fatty acid fermenters such as *Turicibacter* and *Catenibacterium* were significantly higher in samples stored at -20˚C. *Clostridioides hiranonis* which plays a particularly important role in converting primary bile acids to secondary bile acids in dogs, was significantly increased at 6-month and 9-months but decreased at 12 months of storage. In light of these results, it remains to be determined if fecal material from the same donor that differs in bacterial community composition and diversity across storage conditions and duration could lead to differences in clinical efficacy for specific clinical indications of FMT (e.g., FMT for acute enteropathy vs. FMT for adjunctive treatment for obesity or diabetes). Lastly, careful inspection of the distribution of the data of ASVs that significantly differed between time and storage conditions (S1 Fig) shows that for some ASVs (e.g., *Clostridioides hiranonis*) the trend of change across time and storage conditions is not consistent with a meaningful biological pattern. These results might indicate minimal biological significance or a type-1 statistical error.

We demonstrated a variable and progressive decline in the total extracted fecal DNA concentration at the different study's timepoints. At 12-month in both storage conditions, the DNA concentration was lower than baseline. Nevertheless, in storage at -80˚C, the decline in total fecal DNA was not associated with significant changes in alpha diversity, whereas, in storage at -20˚C it was. We contend that fecal DNA extraction and quantification are susceptible to user's pipetting and other method-related errors and that differences in total DNA concentration do not adequately reflect changes in alpha and beta diversity. After considering both the measures of diversity and total fecal DNA content at different storage conditions and duration, we conclude that the storage of 20% glycerol cryopreserved stool at -80˚C was superior to storage at -20˚C.

NMDS plots of the bray-cutis dissimilarity matrix revealed significant separation of samples according to their individual dog of origin, regardless of storage time or condition. However, close examination of the NMDS plots revealed that samples from some individual dogs formed larger, less cohesive clusters. Following up on this observation we calculated beta dispersion statistics (mean distance from centroid) for each sample with respect their dog of origin and compared them to determine if there were differences among samples collected from different dogs. We demonstrated significant differences in beta dispersion among individual dogs during sample storage. This finding suggests that microbiome profiles from individual dogs have variable stability during sample storage with respect to community composition. We then compared the abundance of ASVs between dogs with low and high beta dispersion and identified an unknown species of *Fusobacterium* whose relative abundance was significantly higher in fresh fecal samples from dogs with high beta dispersion, and varied in association with storage at -20˚C. We postulate that certain groups of bacteria may contain cytolytic enzymes that

when released to their near environment could damage other bacteria and lead to drifts in microbiome compositions. In that context, if some dogs have fecal microbiome compositions that are more labile than others, it could impact the abundance of bacteria that could be relevant for FMT's therapeutic mechanisms of action. We think that a similar assessment of a large cohort of dogs is necessary and may assist in the identification of donors with specific compositions of microbial communities that have the potential to destabilize the fecal microbiome under varying time and storage conditions. Interestingly, Shannon alpha diversity indices were higher in the fresh feces of dogs with high beta dispersion during storage. A previous investigation in human FMT recipients revealed a positive association between alpha diversity in the donors' stool and therapeutic efficacy for inflammatory bowel diseases [66, 68], but the impact of donor microbiota diversity on efficacy of treatment for other disorders is not known. Our results suggest that microbiota diversity may have a divergent effect on clinical efficacy and stability. Though high donor alpha diversity is associated with positive responses to FMT therapy, our results suggest it is also associated with greater compositional variation during storage of donor feces. Thus, the alpha diversity prediction may not be parallel between clinical efficacy and product stability. If higher alpha diversity in donor feces is associated with reduced stability during storage, this could impact recommendations for donor selection for FMT products that will be stored for extended periods. Because the relationship between microbiota alpha diversity and the stability of FMT preparations from donor stool is unknown at this time, follow-up investigations are required to confirm and understand the implications of our findings on donor stool screening.

## Supporting information

**S1 Table. Pairwise comparisons of beta dispersion among individual dogs.** Tukey's tests were used to detect statistically significant differences in beta dispersion between pairs of individual dogs.
(CSV)

**S1 Fig. Jitter boxplots of ASVs that differed significantly between time and storage conditions.** The edges of the box represent the 25th and 75th percentiles. The whiskers represent the maximum and minimum values below and above the upper (75th percentile + IQR) and lower (and 25th percentile–IQR) fences, respectively. Individual black dots represent individual dogs.
(PDF)

## Acknowledgments

The authors would like to thank the University of Illinois at Urbana-Champaign Staff and Students dog owners that provided fresh stool samples for this study.

## Author Contributions

**Conceptualization:** Patrick Barko, Arnon Gal.

**Data curation:** Patrick Barko, Arnon Gal.

**Formal analysis:** Patrick Barko, Arnon Gal.

**Funding acquisition:** Arnon Gal.

**Investigation:** Julie Nguyen-Edquilang, Arnon Gal.

**Methodology:** Patrick Barko, Julie Nguyen-Edquilang, Arnon Gal.

**Project administration:** Julie Nguyen-Edquilang, Arnon Gal.

**Resources:** Arnon Gal.

**Software:** Patrick Barko, Arnon Gal.

**Supervision:** Arnon Gal.

**Writing – original draft:** Arnon Gal.

**Writing – review & editing:** Patrick Barko, Julie Nguyen-Edquilang, David A. Williams, Arnon Gal.

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
