## [Decision Letter · Decision Letter 0]

2 Jul 2023

PONE-D-23-13936Fecal microbiome composition and diversity of cryopreserved canine stool at different duration and storage conditionsPLOS ONE

Dear Dr. Gal,

Thank you for submitting your manuscript to PLOS ONE. After careful consideration, we feel that it has merit but does not fully meet PLOS ONE’s publication criteria as it currently stands. Therefore, we invite you to submit a revised version of the manuscript that addresses the points raised during the review process.

Essentially, both reviewers had recommended to provide more details regarding the sample preparation, statistical analysis and bioinformatics used in the methodology. For instance, justification for the filtering criteria should be provided, and the method used to annotate the taxonomy should be described in details to avoid inconsistent nomenclature in the taxonomy. Additionally, the used of PMA should be included in the abstract. 

We look forward to receiving your revised manuscript.

Kind regards,

Chun Wie Chong

Academic Editor

PLOS ONE

Reviewers' comments:

Reviewer's Responses to Questions

**Comments to the Author**

1. Is the manuscript technically sound, and do the data support the conclusions?

Reviewer #1: Yes

Reviewer #2: Partly

2. Has the statistical analysis been performed appropriately and rigorously? 

Reviewer #1: I Don't Know

Reviewer #2: Yes

3. Have the authors made all data underlying the findings in their manuscript fully available?

Reviewer #1: No

Reviewer #2: No

4. Is the manuscript presented in an intelligible fashion and written in standard English?

Reviewer #1: Yes

Reviewer #2: Yes

5. Review Comments to the Author

Reviewer #1: Dear authors, congratulations on a well written article that is contributing to a relevant and exciting area of research. I would like to disclose that my primary area of expertise in fecal and microbiome research is in metabolomics and so I cannot comment on some of the specifics of the analyses performed. That being said, please find my list of questions and comments below:

Questions/comments:

1) Data Availability: you stated "Yes - all data are fully available without restriction" in the Data Availability section, but under the description of 'where the data may be found' you state "Other data presented in this study are available on request". Data availability "on request" is not sufficient according to journal policy to specify a "yes" under the data availability section. Please review the journal policy, correct as needed, and ensure that you are in compliance. Along these same lines it will be important to clearly define what "other data presented" is specifically referring to.

2) Grammar: a) in the Abstract, lines 26 and 27, you state "Certain bacterial taxa had had positive and negative survival correlations", I believe one "had" is sufficient. b) FMT needs to be written out in full in the abstract and not abbreviated. c) Along the same lines, the FMT acronym is used several times prior to it being defined in the manuscript. It is currently defined in the Introduction, lines 50 and 51. This definition and acronym used need to be moved to the first time it is referenced, then the FMT acronym can be used throughout. d) Please define the "MBD" acronym used in Table 1.

3) In the Introduction, line 46 and 47, you state "This specificity allows us to assess changes exclusively in intact bacterial populations." Does "intact" equal "viable"? Do intact bacterial come back to activity? This is addressed in the conclusion, but I think introduction of this premise is important to include in the Introduction.

4) Sample preparation: There are several instances of the discarding of supernatants during sample preparation. First during the preparation of fresh stool, lines 83 and 84, and then again during PMA treatment, lines 105 - 108. In all cases, this step was preceded by a centrifugation step. Bacteria are certainly present in the liquid fraction during these processes, have you confirmed that the centrifugation is sufficient to pellet the bacteria and that you are not losing species in the discarded supernatant fraction? Is there some desire to only characterize bacteria adhered to solid material?

5) Data Filtering: In the Statistical analysis section, lines 172 and 173, and further described in the Results section, lines 228 and 229, you discuss the removal of features from the data. In your analysis you start with over 6 million sequences that got reduced down to just under 4000 sequences. You state that features from unclassified phyla were removed as well as "rare, uninformative features filtered for prevalence and absolute abundance". Given the massive reduction of data and excluded features, how can you be sure you are drawing appropriate conclusions? I would like to see some sort of justification that the removal of those features did not impact conclusions drawn. Also, more visibility in the what fraction of the 6 million were removed and for what specific reason. How does one define "uninformative features"? Why should "unclassified phyla" be removed from the analysis? Their stability profile is potentially just as relevant to successful FMT as classified phyla.

6) Finally, just a comment and not a question, the data you show in Figure 3 aligns very much with the findings from my own research in small molecules. Namely that individual-to-individual differences in feces most frequently exceed storage differences by a strong margin under reasonable storage conditions...ie not stored at RT. An exciting part of this ongoing research will be understanding the impact of the individual-to-individual fecal profile and its impact on successful FMT.

Reviewer #2: The manuscript “Fecal microbiome composition and diversity of cryopreserved canine stool at different duration and storage conditions” describes an interesting and timely work aiming to fill the gap of knowledge in FMT preparation and preservation methods. The study however has some limitations that were not discussed, and several points that concern this reviewer. Also, data was not deposited in a public repository.

Abstract: The abstract did not mention the use of PMA at all, there is no description of materials and methods, glossed over results, and is almost 100% introduction and discussion. Please keep in mind that most people will only read your abstract, so your abstract should be an accurate reflection of your manuscript. Because of limited space, there is no room in the abstract for hypothetical conclusions, and any conclusion should be strictly supported by data. There was no clinical validation on these results and we do not know how any of these changes affect clinical outcomes (more on that later) so please remove or heavily edit your last sentence.

Line 39: while I completely agree that having more viable bacteria is likely a good practice, we do not know whether live bacteria is even necessary, and there is evidence in humans to support that they are not (see Ott et al, Efficacy of Sterile Fecal Filtrate Transfer for Treating Patients With Clostridium difficile Infection, Gastroenterology, Volume 152, Issue 4, 2017, https://doi.org/10.1053/j.gastro.2016.11.010). This needs to be incorporated into your discussion and this statement needs to be rewritten to reflect that.

Lines 45-47: True, but intact doesn’t necessarily mean viable. This is a limitation of the method used that needs to be clarified and discussed.

Lines 87-89: Were baseline samples also homogenized, filtered, and resuspended with glycerol? If yes, they are not a true representation of the viability expected in a fresh sample (which is our gold standard) since there is significant exposure to oxygen which will quickly kill off some of the strict anaerobes; if no, then the processing of the samples is adding an additional layer of loss of viability (oxygen exposure plus freezing). Ideally both should have been performed, but since it cannot be re-done, this needs to be clearly explained in the materials and methods, and thoroughly discussed in the study limitations.

Lines 138-151: I commend the authors for adding negative controls to the project, however, they need to be analyzed beyond simply stating what was found in them (especially true in this case since you found some bacteria that were reported later as significant). I would suggest using the decontam script (Davis, N.M., Proctor, D.M., Holmes, S.P. et al. Simple statistical identification and removal of contaminant sequences in marker-gene and metagenomics data. Microbiome 6, 226 (2018). https://doi.org/10.1186/s40168-018-0605-2) to remove all contaminants from the samples before proceeding with the analysis.

Lines 229-231: Phyla names have recently changes, and while it is still ok to use the old names, you are reporting a mix and match of old and new names. Please be consistent and use only old or only new names.

Table 2: Please include baseline mean and SE for comparison. I personally dislike this table as it is highlighting the differences between protocols rather than the differences to baseline which are much more important for the clinical application. It is misleading to show significance at 3 months between the protocols when neither was significantly different from baseline (but this information is buried in the text rather than highlighted in a table). If you could incorporate both analysis into the table it would be a significant improvement.

Table 3: Replace fresh with baseline.

Table 4: The Clostridium hiranonis data popped up to my eyes because of its relevance but my comment here probably applies to the other bacteria as well: how can you explain that hiranonis was positively affected by freezing from 3 to 6/9 months, but then suddenly it was negatively affected by freezing at 9-12 and 6-12 months? Biologically speaking, it doesn’t make sense. I could understand no difference then negatively affected, but not switching like this. It suggests to me that either the effect is biologically minimal and the significance is a statistical artifact, or that there is a mistake in your data. This may also change once you remove contaminants from your dataset (hopefully that is the case). In any case, please clarify and add scatterplot graphs so we can look at the distribution of the data and see if you are being driven by outliers or if this is a consistent effect across all samples.

Figure 1: please add baseline to the figure.

Lines 368-369: Again, viable and intact are not synonyms, please correct across the manuscript.

Lines 380-390: See comment on table 4. The statement at the end of this paragraph is made on a very shaky premise as we do not know if A) viable bacteria are even necessary for the success of FMT (see Ott 2017 for evidence against it); or B) whether the changes in viability observed here are clinically relevant (as you could have a significant decrease but still have plenty of bacteria to replenish a bacterial pathway or for recolonization). Given the large variability of how much fecal material is given in different published FMT studies (especially if you try to calculate in grams of feces per Kg of body weight) it is hard to say how much of each bacteria is truly needed and how suboptimal abundance/viability affect success rates or outcomes. Give the small size effect in alpha and beta diversity you may still have enough for a clinical effect but without an actual clinical trial you cannot draw any conclusions here on clinical effect. Please refrain from making any statements into clinical indications as those are purely speculations at this stage.

Lines 403-421: On your beta diversity graphs we can see a bigger effect on beta diversity for individual animals (dog effect) than the effect of the conditions tested. I think your data hints more about the importance of donor selection vs the preservation method – which makes sense since you only tested optimal strategies with glycerol and there was no control without a cryoprotectant. I would like to see the dog effect calculated on your dataset to compare the size effect between the individual variation and the effect of preservation.

6. PLOS authors have the option to publish the peer review history of their article (what does this mean?). If published, this will include your full peer review and any attached files.

Reviewer #1: No

Reviewer #2: No

---

## [Author Response · Author response to Decision Letter 0]

24 Aug 2023

Dear Editor and Reviewers,

We thank the Reviewers for their thoughtful critique and for the time spent on reviewing our manuscript and providing comments to improve it. In the revised manuscript we addressed all of the Reviewers’ comments and made the majority of suggested revisions, whereas, we address few others in this response letter. These modifications have strengthened our manuscript, and we are optimistic they will satisfy the Reviewers’ concerns. Please see below our line-by-line responses. All manuscript changes were made using the Microsoft’s Track-Change function as instructed by the Editor.

Authors’ response: To the best of our ability, we think that our manuscript conforms with PLOS ONE guidelines.

Authors’ response: We apologize for this glitch that has now been fixed. Both the R code and the raw sequences are now present in their respective repositories.

Reviewer #1: 

Dear authors, congratulations on a well written article that is contributing to a relevant and exciting area of research. I would like to disclose that my primary area of expertise in fecal and microbiome research is in metabolomics and so I cannot comment on some of the specifics of the analyses performed. That being said, please find my list of questions and comments below:

1) Data Availability: you stated "Yes - all data are fully available without restriction" in the Data Availability section, but under the description of 'where the data may be found' you state "Other data presented in this study are available on request". Data availability "on request" is not sufficient according to journal policy to specify a "yes" under the data availability section. Please review the journal policy, correct as needed, and ensure that you are in compliance. Along these same lines it will be important to clearly define what "other data presented" is specifically referring to.

Authors’ response: We apologize for this glitch that has now been fixed. Both the R code and the raw sequences are now present in their respective repositories. 

2) Grammar: a) in the Abstract, lines 26 and 27, you state "Certain bacterial taxa had had positive and negative survival correlations", I believe one "had" is sufficient. b) FMT needs to be written out in full in the abstract and not abbreviated. c) Along the same lines, the FMT acronym is used several times prior to it being defined in the manuscript. It is currently defined in the Introduction, lines 50 and 51. This definition and acronym used need to be moved to the first time it is referenced, then the FMT acronym can be used throughout. d) Please define the "MBD" acronym used in Table 1.

Authors’ response: We revised the abstract, introduction and Table 1 according to the Reviewer suggestion.

3) In the Introduction, line 46 and 47, you state "This specificity allows us to assess changes exclusively in intact bacterial populations." Does "intact" equal "viable"? Do intact bacterial come back to activity? This is addressed in the conclusion, but I think introduction of this premise is important to include in the Introduction.

Authors’ response: We revised the introduction to include a statement that intact bacterial cells are not necessarily viable or functional (line 65) and that there is some evidence that FMT may be efficacious when transplanting bacterial cellular components from dead bacteria (lines 46-49).

4) Sample preparation: There are several instances of the discarding of supernatants during sample preparation. First during the preparation of fresh stool, lines 83 and 84, and then again during PMA treatment, lines 105 - 108. In all cases, this step was preceded by a centrifugation step. Bacteria are certainly present in the liquid fraction during these processes, have you confirmed that the centrifugation is sufficient to pellet the bacteria and that you are not losing species in the discarded supernatant fraction? Is there some desire to only characterize bacteria adhered to solid material?

Authors’ response: The Reviewer raises an important point. The first step of centrifugation described in the revised manuscript in line 103 was necessary to generate a pellet that would be resuspended with the cryopreservative. The additional steps of centrifugation described in line 125 were made to discard residues of PMA that had the potential to affect downstream DNA amplification. This is why we chose to perform it this way. We agree that there might have been some loss of bacteria that were not pelleted and remained in the supernatant or of strict anaerobes that were exposed to oxygen. Unfortunately, we did not check it, so we do not know how much was lost during these steps. We did however generate 6,004,861 long-read 16S-rRNA amplicons so we postulate that if something was lost it may not have been in significant amounts. Since there is a likelihood that the Reviewer is correct, we added to the discussion a comment about this potential shortcoming (lines 422-427). 

5) Data Filtering: In the Statistical analysis section, lines 172 and 173, and further described in the Results section, lines 228 and 229, you discuss the removal of features from the data. In your analysis you start with over 6 million sequences that got reduced down to just under 4000 sequences. You state that features from unclassified phyla were removed as well as "rare, uninformative features filtered for prevalence and absolute abundance". Given the massive reduction of data and excluded features, how can you be sure you are drawing appropriate conclusions? I would like to see some sort of justification that the removal of those features did not impact conclusions drawn. Also, more visibility in the what fraction of the 6 million were removed and for what specific reason. How does one define "uninformative features"? Why should "unclassified phyla" be removed from the analysis? Their stability profile is potentially just as relevant to successful FMT as classified phyla.

Authors’ response: We regret that our description of the sequencing results has resulted in a misunderstanding of our findings. The manuscript has been revised for clarity and the term “uninformative” has been removed (lines 200-203). Sequencing of the full-length 16S-rRNA amplicons in 189 samples generated over 6 million sequencing reads. These raw sequences were binned into 3,582 amplicon sequence variants (ASVs), each containing reads with identical sequences. No fraction of these 6 million reads were discarded and the subsequent filtering steps were used to removed rare ASVs that were present in a small number of samples. Thus, we did not reduce the number of sequences from 6 million to 3,582, we grouped identical sequences into 3,582 ASVs. This is a typical number of ASVs for investigations of mammalian fecal microbiomes. Fecal microbiome datasets are also sparse and contain hundreds of relatively rare ASVs with low prevalence among samples. These ASVs are uninformative because they are often identified in a single sample or a very small number of samples. Statistical analysis of these sequences for comparisons among groups are not valid or useful due to the very low prevalence of the ASVs. Owing to the sparsity of microbiome feature tables, it is common for hundreds-thousands of features to be removed during this filtering step. Including these uninformative features for differential abundance analysis also inflates the risk of false discovery due to making 3,582 individual statistical comparisons (one for every ASV). For these reasons, rare features with low prevalence are removed from the dataset prior to ordination and differential abundance analysis. Conversely, the presence of rare ASVs is important for analyses of microbial diversity. We used the unfiltered dataset for alpha diversity analysis and the filtered data for ordination (beta diversity) and differential abundance analysis. Regarding removal of undefined phyla, this was a judgement made by the investigators to exclude from analysis features about which critical taxonomic data was lacking. Full-length 16S-rRNA amplicon sequencing was used in this study to generate feature tables with taxonomic resolution at the genus and species level and the data were agglomerated at the taxonomic level of species. As we could not identify features with unknown phyla with any confidence, we considered them uninformative and removed them. 

6) Finally, just a comment and not a question, the data you show in Figure 3 aligns very much with the findings from my own research in small molecules. Namely that individual-to-individual differences in feces most frequently exceed storage differences by a strong margin under reasonable storage conditions...ie not stored at RT. An exciting part of this ongoing research will be understanding the impact of the individual-to-individual fecal profile and its impact on successful FMT.

Authors’ response: We completely agree with the Reviewer. 

Reviewer #2: 

The manuscript “Fecal microbiome composition and diversity of cryopreserved canine stool at different duration and storage conditions” describes an interesting and timely work aiming to fill the gap of knowledge in FMT preparation and preservation methods. The study however has some limitations that were not discussed, and several points that concern this reviewer. Also, data was not deposited in a public repository.

Authors’ response: We apologize for this glitch that has now been fixed. Both the R code and the raw sequences are now present in their respective repositories. 

Abstract: The abstract did not mention the use of PMA at all, there is no description of materials and methods, glossed over results, and is almost 100% introduction and discussion. Please keep in mind that most people will only read your abstract, so your abstract should be an accurate reflection of your manuscript. Because of limited space, there is no room in the abstract for hypothetical conclusions, and any conclusion should be strictly supported by data. There was no clinical validation on these results and we do not know how any of these changes affect clinical outcomes (more on that later) so please remove or heavily edit your last sentence.

Authors’ response: The abstract was revised as recommended.

Line 39: while I completely agree that having more viable bacteria is likely a good practice, we do not know whether live bacteria is even necessary, and there is evidence in humans to support that they are not (see Ott et al, Efficacy of Sterile Fecal Filtrate Transfer for Treating Patients With Clostridium difficile Infection, Gastroenterology, Volume 152, Issue 4, 2017, https://doi.org/10.1053/j.gastro.2016.11.010). This needs to be incorporated into your discussion and this statement needs to be rewritten to reflect that.

Authors’ response: The Reviewer raised an important and a very interesting argument. We postulate that the clinical indication for FMT and the duration of effect are likely going to determine if the fecal material should contain viable and functional bacteria OR if bacterial cell components would suffice. For example, if biotransformation of secondary bile acids from primary bile acids is sought, it is not very likely that bacterial cell extracts will be enough to achieve a lasting effect, particularly if a single FMT would be performed. We incorporated the reviewer comment and reference in the introduction and expanded on this limitation in the discussion (lines 418-422).

Lines 45-47: True, but intact doesn’t necessarily mean viable. This is a limitation of the method used that needs to be clarified and discussed.

Authors’ response: We agree with the Reviewer and have revised the introduction to include a statement that intact bacterial cells are not necessarily viable or functional and that there is some evidence that FMT may be efficacious when transplanting bacterial cellular components from dead bacteria.

Lines 87-89: Were baseline samples also homogenized, filtered, and resuspended with glycerol? If yes, they are not a true representation of the viability expected in a fresh sample (which is our gold standard) since there is significant exposure to oxygen which will quickly kill off some of the strict anaerobes; if no, then the processing of the samples is adding an additional layer of loss of viability (oxygen exposure plus freezing). Ideally both should have been performed, but since it cannot be re-done, this needs to be clearly explained in the materials and methods, and thoroughly discussed in the study limitations.

Authors’ response: With respect, we request that the Reviewer consider the study hypothesis and purpose in the context of their comment. This was a preclinical study aiming to determine what is the effect of storage time and temperature on cryopreserved donor material. The intent is to generate results and further hypotheses that would inform clinical practices of preservation of canine donor material for FMT in real-life. In this particular context, it is our opinion that it is more important to compare the cryopreserved treatment group samples to cryopreserved baseline samples than to fresh non-cryopreserved samples because in real-life scenario the donor samples will inevitably be treated for cryopreservation prior to storage and as the reviewer pointed out, this step will lead to some loss of strict anaerobes. Hence in the context of assessing the effect of time and storage on cryopreserved stool, it would make less sense to compare the cryopreserved treatment group samples to fresh non-cryopreserved samples. 

Not to discount the Reviewer’s opinion, we have included a comment in the limitation paragraph in the discussion section indicating that centrifugation may have led to loss of strict anaerobes (lines 422-427). 

Lines 138-151: I commend the authors for adding negative controls to the project, however, they need to be analyzed beyond simply stating what was found in them (especially true in this case since you found some bacteria that were reported later as significant). I would suggest using the decontam script (Davis, N.M., Proctor, D.M., Holmes, S.P. et al. Simple statistical identification and removal of contaminant sequences in marker-gene and metagenomics data. Microbiome 6, 226 (2018). https://doi.org/10.1186/s40168-018-0605-2) to remove all contaminants from the samples before proceeding with the analysis.

Authors’ response: We thank the Reviewer for this important comment and helpful suggestion. As recommended, we have used the “decontam” script on our data and identified two contaminating ASVs in the negative control samples. These ASVs were removed, and the data were reanalyzed. The manuscript has been revised to reflect the updated methods (lines 195-197) and results (lines 260-273) as necessary. 

Lines 229-231: Phyla names have recently changes, and while it is still ok to use the old names, you are reporting a mix and match of old and new names. Please be consistent and use only old or only new names.

Authors’ response: We are unsure of which taxonomic identifiers the Reviewer considered to be “old” vs. “new” but we have described the two methods of taxonomic assignment in the “Materials and Methods.” We used the most up-to-date versions of the respective databases available when conducting our analyses. The discrepancy in taxonomic identifiers could have emerged from conflicts between the SILVA-based identifiers and those in the Shoreline Athena databased which are derived from genomes in the NCBI database. The Shoreline Athena databases is thought to be more specific at the level of species and sub-species/strain. Owing to some differences in the taxonomic annotations between the SILVA and the Shoreline Athena databases we present the taxonomic annotations from both databases in Tables 3 and 4 and made a comment about those taxonomic differences in the method section (lines 171-177).

Table 2: Please include baseline mean and SE for comparison. I personally dislike this table as it is highlighting the differences between protocols rather than the differences to baseline which are much more important for the clinical application. It is misleading to show significance at 3 months between the protocols when neither was significantly different from baseline (but this information is buried in the text rather than highlighted in a table). If you could incorporate both analysis into the table it would be a significant improvement.

Authors’ response: We added the baseline values in Table 2.

Table 3: Replace fresh with baseline.

Authors’ response: the requested changes were made.

Table 4: The Clostridium hiranonis data popped up to my eyes because of its relevance but my comment here probably applies to the other bacteria as well: how can you explain that hiranonis was positively affected by freezing from 3 to 6/9 months, but then suddenly it was negatively affected by freezing at 9-12 and 6-12 months? Biologically speaking, it doesn’t make sense. I could understand no difference then negatively affected, but not switching like this. It suggests to me that either the effect is biologically minimal and the significance is a statistical artifact, or that there is a mistake in your data. This may also change once you remove contaminants from your dataset (hopefully that is the case). In any case, please clarify and add scatterplot graphs so we can look at the distribution of the data and see if you are being driven by outliers or if this is a consistent effect across all samples.

Authors’ response: Thank you for this comment. We reanalyzed the data after running the decnotam script and the results remained the same. The revised manuscript now also contains the Sup1 Figure (as recommended by the Reviewer) which consists of jitter boxplots of all the ASVs that significantly differed between timepoints and storage conditions so that the readers and the Reviewer could see the distribution of the data and make their own opinion on the significance of the statistical analysis results and its biological meaning (including the case of Clostridium hiranonis). We agree with the Reviewer that in the case of Clostridium hiranonis it is most likely a statistical artifact and we made a comment about it in the discussion (lines 439-443).

Figure 1: please add baseline to the figure.

Authors’ response: Figure 1B contains the mean (±SD) baseline information that the reviewer requested. Figure 1A plots the Δ mean (±SE) log fecal DNA concentration from baseline (i.e., baseline log DNA - timepoint log DNA) in -20°C and -80°C storage. Given that we plotted the difference from baseline, the actual baseline is not in the plot as a stand-alone. We think that plotting the trend of the difference from baseline is more informative when trying to demonstrate the effect of storage on DNA content.

Lines 368-369: Again, viable and intact are not synonyms, please correct across the manuscript.

Authors’ response: The requested change was made.

Lines 380-390: See comment on table 4. The statement at the end of this paragraph is made on a very shaky premise as we do not know if A) viable bacteria are even necessary for the success of FMT (see Ott 2017 for evidence against it); or B) whether the changes in viability observed here are clinically relevant (as you could have a significant decrease but still have plenty of bacteria to replenish a bacterial pathway or for recolonization). Given the large variability of how much fecal material is given in different published FMT studies (especially if you try to calculate in grams of feces per Kg of body weight) it is hard to say how much of each bacteria is truly needed and how suboptimal abundance/viability affect success rates or outcomes. Give the small size effect in alpha and beta diversity you may still have enough for a clinical effect but without an actual clinical trial you cannot draw any conclusions here on clinical effect. Please refrain from making any statements into clinical indications as those are purely speculations at this stage.

Authors’ response: With respect to the Reviewer, the discussion section is the appropriate place to provide the authors’ opinions and interpretation on their data. The Reviewer’s arguments about 1) bacterial cellular extracts, and 2) whether the changes in viability are clinically relevant are sound but do not preclude the possibility that changes in beta-diversity could lead to differences in efficacy for specific clinical indications. It is well accepted that the main driving force in beta diversity is the donor factor predominantly because of inter-donor differences in beta-diversity and that the latter has a strong impact on FMT efficacy. We also demonstrated a strong donor effect in this study that is augmented by a much weaker effect of storage, yet even being small compared with the larger dog effect, it may still be significant and neither the Reviewer nor us know the answer to that question until it is put to an experiment. Therefore, we toned down our statement, but we did not retract it (lines 435-439).

Lines 403-421: On your beta diversity graphs we can see a bigger effect on beta diversity for individual animals (dog effect) than the effect of the conditions tested. I think your data hints more about the importance of donor selection vs the preservation method – which makes sense since you only tested optimal strategies with glycerol and there was no control without a cryoprotectant. I would like to see the dog effect calculated on your dataset to compare the size effect between the individual variation and the effect of preservation.

Authors’ response: In the paragraph that comes immediately following Table 2, we addressed the reviewer concern regarding the dog effect compared to the storage effect by specifying the PERMANOVA F-statistic and p values for 1) individual dogs, 2) storage temperature, and 3) time (lines 297-304).

---

## [Decision Letter · Decision Letter 1]

28 Sep 2023

PONE-D-23-13936R1Fecal microbiome composition and diversity of cryopreserved canine stool at different duration and storage conditionsPLOS ONE

Dear Dr. Gal,

Thank you for submitting your manuscript to PLOS ONE. After careful consideration, we feel that it has merit but does not fully meet PLOS ONE’s publication criteria as it currently stands. Therefore, we invite you to submit a revised version of the manuscript that addresses the points raised during the review process.

We look forward to receiving your revised manuscript.

Kind regards,

Chun Wie Chong

Academic Editor

PLOS ONE

Journal Requirements:

Reviewers' comments:

Reviewer's Responses to Questions

**Comments to the Author**

1. If the authors have adequately addressed your comments raised in a previous round of review and you feel that this manuscript is now acceptable for publication, you may indicate that here to bypass the “Comments to the Author” section, enter your conflict of interest statement in the “Confidential to Editor” section, and submit your "Accept" recommendation.

Reviewer #2: All comments have been addressed

Reviewer #3: (No Response)

2. Is the manuscript technically sound, and do the data support the conclusions?

Reviewer #2: Yes

Reviewer #3: Partly

3. Has the statistical analysis been performed appropriately and rigorously? 

Reviewer #2: Yes

Reviewer #3: I Don't Know

4. Have the authors made all data underlying the findings in their manuscript fully available?

Reviewer #2: Yes

Reviewer #3: Yes

5. Is the manuscript presented in an intelligible fashion and written in standard English?

Reviewer #2: Yes

Reviewer #3: Yes

6. Review Comments to the Author

Reviewer #2: (No Response)

Reviewer #3: Line 50: Authors to update naming nomenclature for "Clostridum" to "Clostridioides" throughout the article and note that recurrent CDI-associated colitis is different from recurrent CDI.

Line 214: Microbial diversity and composition are largely influenced by diet, and the lack of diet log variation amongst these dogs had and environmental microbes from soil that they were exposed to was missing from the manuscript as a limitation to analysis. If such data is available, propose to cluster dogs by type of food diet and area they were exposed to for clustering in microbial composition and diversity rather than via breed to ascertain which composition is more stable by storage condition and duration.

7. PLOS authors have the option to publish the peer review history of their article (what does this mean?). If published, this will include your full peer review and any attached files.

Reviewer #2: **Yes: **Rachel Pilla

Reviewer #3: No

---

## [Author Response · Author response to Decision Letter 1]

3 Oct 2023

Reviewer #3: 

Line 50: Authors to update naming nomenclature for "Clostridum" to "Clostridioides" throughout the article and note that recurrent CDI-associated colitis is different from recurrent CDI.

Authors’ response: Per the Reviewer’s suggestion we changed "Clostridum" to "Clostridioides" throughout the manuscript. 

Also, we have used the term “recurrent CDI-associated colitis” interchangeably with “recurrent CDI”. We see the Reviewer’s point that recurrent CDI-associated colitis is a colonic manifestation of recurrent CDI and because what we had previously referred to in the manuscript was recurrent CDI-associated colitis, in this revised manuscript we changed “recurrent CDI” to “recurrent CDI-associated colitis” throughout the manuscript.

Line 214: Microbial diversity and composition are largely influenced by diet, and the lack of diet log variation amongst these dogs had and environmental microbes from soil that they were exposed to was missing from the manuscript as a limitation to analysis. If such data is available, propose to cluster dogs by type of food diet and area they were exposed to for clustering in microbial composition and diversity rather than via breed to ascertain which composition is more stable by storage condition and duration.

Authors’ response: We did not collect those data so unfortunately, we cannot follow the Reviewer’s suggestion. We added the following sentence in lines 428-432 to highlight the Reviewer’s comment: “Lastly, we did not record the diet and geographical locations where the dogs resided. Both diet and soil bacteria from different geographic locations could have impacted the gut microbial community structure of the dogs. Including these factors in the multivariate analysis potentially could have revealed important information about their effect on the stability of cryopreserved stool.”

---

## [Editor Report · Decision Letter 2]

8 Nov 2023

Fecal microbiome composition and diversity of cryopreserved canine stool at different duration and storage conditions

PONE-D-23-13936R2

Dear Dr. Gal,

We’re pleased to inform you that your manuscript has been judged scientifically suitable for publication and will be formally accepted for publication once it meets all outstanding technical requirements.

Kind regards,

Chun Wie Chong

Academic Editor

PLOS ONE
---

## [Editor Report · Acceptance letter]

10 Nov 2023

PONE-D-23-13936R2 

Fecal microbiome composition and diversity of cryopreserved canine stool at different duration and storage conditions 

Dear Dr. Gal:

I'm pleased to inform you that your manuscript has been deemed suitable for publication in PLOS ONE. Congratulations! Your manuscript is now with our production department. 

Kind regards, 

on behalf of

Dr. Chun Wie Chong 

Academic Editor

PLOS ONE